# Impact of a Nyakaza Move-for-Health Intervention Programme among Adolescents in a Resource-Constrained South African Community

**DOI:** 10.3390/ijerph21060717

**Published:** 2024-05-31

**Authors:** Patrick Mkhanyiseli Zimu, Hendrik Johannes van Heerden, Jeanne Martin Grace

**Affiliations:** School of Health Sciences, College of Health Sciences, University of KwaZulu-Natal, Westville Campus, Durban 4000, South Africa; vanheerdenj@ukzn.ac.za (H.J.v.H.); gracej@ukzn.ac.za (J.M.G.)

**Keywords:** community-based, health promotion, non-communicable disease, physical inactivity

## Abstract

Adolescents in resource-constrained environments face increasing sedentary lifestyles and obesity rates, necessitating effective interventions for promoting physical activity and combating non-communicable diseases. This study evaluates the impact of a 12 week Nyakaza Move-for-Health intervention on physical activity, anthropometry, cardiorespiratory fitness, and behaviour change among adolescents in KwaZulu-Natal, South Africa. One hundred twenty-eight adolescents participated, with assessments including self-reported physical activity, anthropometric indices, and cardiorespiratory fitness measures. The intervention significantly increased physical activity levels. The treatment group’s mean score was 2.2 (0.4) at baseline and 2.6 (0.4) (F (14, 79) = 4.884, *p* = <0.001, η2 = 0.156) at the endline. The body mass index decreased (mean = 22.4 (4.6) at baseline and 21.9 (4.2) at endline; *p* = 0.025, partial eta squared = 0.025, η2 = 0.004). The intervention programme significantly affected the WHR (mean = 0.80 (0.10) at baseline and 0.76 (0.9) at endline; *p* < 0.001, partial eta squared = 0.327, η2 = 0.100) and the predicted maximal oxygen uptake (VO_2_ max) for the treatment group (mean = 42.4 (8.7) at baseline and mean = 43.6 (8.7) at endline; *p* < 0.711, partial eta squared = 0.017, η2 = 0.033). Focus group discussions indicated shifts in knowledge, attitudes, and motivation towards physical activity. Power analysis revealed strong observed power (PA: 0.983, BMI: 0.098, WHR: 0.887), indicating the robustness of the intervention’s effects. These findings underscore the effectiveness of the intervention in improving physical health outcomes. It is recommended that longitudinal studies be conducted to assess the long-term sustainability and impact of such interventions on adolescents’ health outcomes, thereby informing the development of comprehensive public health policies and programmes to promote physical activity and combat non-communicable diseases in similar settings.

## 1. Introduction

Non-communicable diseases (NCDs), including cardiovascular diseases (CVDs), cancers, chronic respiratory diseases, and diabetes, account for 74% of all attributable deaths globally [1]. The significant impact of NCDs is further emphasised by the fact that approximately 41% of global mortality among adolescent boys and 48% among adolescent girls has been attributed to NCDs and their risk factors [2]. The World Health Organization has recommended strategies commonly known as “best buys” which could be used to prevent and manage NCDs and their risk factors, including promoting physical activity [3]. Although physical activity interventions have been proven to positively affect health by increasing daily energy expenditure [4], improving health-related physical fitness [5,6], and enhancing overall health and wellness, it is concerning that the physical activity profile of 30–40% of South African children and adolescents is considered to be insufficient [7]. South African boys (20%) and girls (32%) are reportedly overweight, with 9% of boys and 13% of girls considered obese [8,9].

Additionally, approximately 50% of South African adolescents are reported to be sedentary [7]. These findings suggest a concerning susceptibility of South African adolescents to NCDs, either now or later in life [10,11], due to the reported risk factors associated with NCDs. To effectively address being overweight, obesity, and other risks of NCDs among adolescents, proactive interventions are imperative. The existing literature highlights the socioeconomic disparities in adolescents’ physical activity levels, revealing that those from a higher socioeconomic status (HSES) engage more in moderate-to-vigorous physical activity (MVPA) in school and club sports. In contrast, their counterparts from low socio-economic status (LSES) communities spend less time in MVPA within similar settings and communities [12,13].

Consequently, recognizing the need to introduce targeted interventions in low-resource communities, such as the Nyakaza Move-for-Health programme, becomes particularly pertinent. This South African initiative recognises the challenges faced by adolescents in resource-constrained settings. The Nyakaza intervention adopts a comprehensive approach rooted in the transtheoretical model (TTM) [14,15], Health Belief Model (HBM) [16], and human movement conceptual framework [17]. The TTM posits that behaviour change is a process that goes through different stages of change (pre-contemplation, contemplation, preparation, action, and maintenance). Movement across the stage of the change continuum requires self-re-evaluation, which can be initiated through the application of HBM constructs (perceived susceptibility, perceived severity, benefits, and barriers) to mitigate negative health behaviour [16]. Moving across the stage of change and ultimately engaging in physical activity supports the human movement conceptual framework, which highlights the significance of physical activity (PA) behaviour in enhancing physical and physiological outcomes, thereby improving quality of life.

The personal, behavioural, and environmental determinants of physical activity among adolescents include self-efficacy, attitude towards physical activity, the enjoyment of physical activity, exercise history, skills, access, cost, time barriers, and social and cultural support [11,17,18,19,20,21,22]. These factors must be considered in the design of effective population-level intervention programmes. Therefore, the Nyakaza intervention was developed using an intervention mapping framework [23] which allows for the identification of individual, behavioural, and environmental determinants of physical activity among adolescents. 

Through its unique components, this research seeks to contribute to the effectiveness of behaviour change interventions, aligning with the critical call for tailored programmes in underprivileged settings. Within the context described above, this study aims to evaluate the Nyakaza programme’s impact on altering physical activity behaviour, improving physical activity levels, and modifying anthropometric characteristics among adolescents in resource-constrained communities.

## 2. Materials and Methods

### 2.1. Study Design and Population

The study employed a quasi-experimental research design to evaluate the effects of a 12 week physical activity programme on physical activity behaviour, fitness levels, and anthropometric characteristics among adolescents. The study occurred in Clermont and Mpumalanga townships within the eThekwini Metropolitan Municipality in KwaZulu-Natal Province, South Africa. Participants were selected from 353 adolescents already engaged in the Nyakaza project, who were drawn from eight randomly chosen secondary schools. In the first phase of the Nyakaza project, which involved 353 secondary school adolescent learners across eight schools, physical activity levels were assessed to identify those who were insufficiently physically active. Adolescents with a physical activity score of less than 2.75 on the Physical Activity Questionnaire for Adolescents (PAQ-A) [24] were deemed insufficiently physically active and were eligible for participation in the current study. Additionally, the selected participants completed a stages of change questionnaire, where 62% were at the pre-contemplation stage and 38% were at the contemplation stage [14,15]. Consequently, 129 adolescents meeting the inclusion criteria were recruited, with 85 adolescent learners assigned to the treatment group and 44 to the control group.

The real-world setting (school and community) in which this study took place lends itself strongly to using a quasi-experimental design as opposed to a true randomised controlled trial (RCT). The ecological validity of the study was enhanced by the quasi-experimental design. Conducting research in naturalistic environments such as schools and communities allows for the examination of interventions in settings where they are likely to be implemented in practice. This approach helps to capture the complexities and variabilities inherent in real-world settings, providing insights that are more generalisable and applicable to similar contexts.

### 2.2. Data Collection Procedure

Baseline and endline measurements, encompassing physical activity levels, and anthropometric measurements (height, body mass, BMI, waist-to-hip circumference ratio (WHR), and cardiorespiratory fitness (predicted VO_2_ max)) were collected from participants in both the treatment and control groups. Anthropometry and cardiorespiratory fitness tests followed a standardised test battery for Assessing Levels of Physical Activity (ALPHA) in children and adolescents [25]. Physical activity levels were subjectively assessed using the Physical Activity Questionnaire for Adolescents (PAQ-A), which employs a seven-day recall and provides general estimates of physical activity levels for high school learners in grades 9–12 [24]. The PAQ-A has been tested for content and construct validity in similar populations in African and European studies [26,27,28,29]. Through correlations between the PAQ-A scores and objective measures using motion sensors such as accelerometers, the PAQ-A has reported correlations ranging from moderate to high, indicating good alignment between the questionnaire items and the underlying constructs of interest. Test-retest reliability scores from the validation studies conducted have consistently demonstrated high levels of stability and consistency in adolescents’ responses over time, bolstering confidence in the instrument’s reliability for longitudinal assessment [26,27,28,29]. Studies evaluating the reliability of the PAQ-A have reported intraclass correlation coefficients (ICCs) ranging from approximately 0.60 to 0.80 [28]. Participants’ cardiorespiratory fitness was assessed using a standardised 20 metre multistage fitness test [25]. South African validation studies have corroborated the beep test’s criterion-related validity against gold standard measures such as the VO_2_ max alongside demonstrating high test-retest reliability, further validating its use in the study [4,30]. This measure was chosen for cardiorespiratory assessment due to its practicality and suitability for use in rural and township communities. Additionally, a focus group discussion involving a randomly selected group of 48 participants, with 12 from each of the four participating schools, was conducted to examine the psychosocial impact of the programme. By employing a quasi-experimental design and utilizing well-validated assessment tools, this study aimed to balance ecological validity and methodological rigor, ensuring that the findings were both relevant to real-world settings and robust in their conclusions.

### 2.3. Intervention Programme Components and Procedure

The Nyakaza intervention programme represents a comprehensive and theoretically grounded initiative for addressing diverse physical activity facets among at-risk adolescents. The intervention programme comprises two synergistically arranged components: a social marketing campaign and an afterschool PA session. The intervention programme was facilitated by postgraduate students with degrees in sport science who were deemed qualified for their roles. These individuals received specialised training and possessed the necessary expertise to effectively implement the intervention components. Additionally, trained community and healthy lifestyle coordinators were also involved in facilitating the programme. 

#### 2.3.1. Social Marketing Campaign (Component 1)

The Nyakaza social marketing campaign was designed to promote physical activity among adolescents by integrating behaviour change theories and feedback from focus group discussions collected in phase one of the Nyakaza project. Drawing upon the transtheoretical model (TTM) [15] and Health Belief Model (HBM) [16], the campaign emphasises the importance of physical activity and raises awareness about the risks of a sedentary lifestyle and non-communicable diseases. Communication materials designed to cater to the unique characteristics and preferences of the adolescents were disseminated through various channels such as infographics, posters, and diaries, addressing personal risks associated with physical inactivity. Prior insights from focus group discussions informed the campaign’s design and distribution, ensuring relevance and effectiveness. Campaign materials were strategically placed in schools to maximise exposure, reflecting the participants’ preferences. The participants actively engaged with the campaign materials, incorporating them into their daily routines and participating in discussions to reflect on the messages and their impact on behaviour change. Overall, the campaign aimed to empower adolescents to adopt and maintain active lifestyles, contributing to the prevention of non-communicable diseases in the community.

#### 2.3.2. Afterschool PA Session (Component 2)

Component 2 of the Nyakaza intervention, the afterschool PA programme, represents a critical stage in facilitating adolescent behaviour change, building upon the awareness and motivation initiated in Component 1. While Component 1 focused on raising awareness and initiating the behaviour change process, Component 2 took this further by providing structured opportunities for adolescents to engage in physical activity. This aligns with the “action” stage of the transtheoretical model (TTM), where individuals are motivated to make tangible changes in their behaviour. The structured cardio dance classes and muscular endurance circuit training sessions offered in Component 2 provide adolescents with the means to actively participate in physical activity, addressing the need for structured opportunities highlighted in Component 1. The programme’s design, frequency, duration, and intensity were tailored to gradually increase participants’ engagement and fitness levels over the 12 week intervention period, aligning with the principles of behaviour change and physical fitness improvement. Furthermore, the involvement of trained community healthy lifestyle facilitators, school educators, and research assistants ensured the effective implementation and monitoring of the programme, enhancing its potential impact on participants’ physical activity levels and overall health. Overall, component 2 complements Component 1 by providing the necessary infrastructure and support for adolescents to translate their awareness and motivation into meaningful behaviour change, ultimately contributing to preventing non-communicable diseases in the community.

Table 1 outlines the intervention programme’s change objectives, performance objectives, methods, and practical applications. The methodological approach integrates social marketing strategies, conscious-raising messages and persuasive communication, practical applications like exposing participants to campaign materials, and a 12 week physical activity programme. Change objectives and methods were delineated for both groups. Change objectives 1 and 2 were for both the treatment and control groups, while change objectives 3 and 4 pertained solely to the treatment group.

### 2.4. Data Analysis

Quantitative and qualitative methods were applied sequentially (QUAN + QUAL) [31]. The collected quantitative and qualitative data were analyzed separately but later merged to interpret results comprehensively. A factorial ANOVA test was used for quantitative data to determine the significant main effects of each independent variable and the interactions between these factors with the dependent variables. Post hoc power analysis was conducted to determine whether the sample size used in the research was sufficient to detect significant effects, if they existed, with a reasonable degree of confidence. An alpha level of 0.05 was set at the threshold for statistical significance, meaning that a result was considered statistically significant if the *p* value was less than or equal to 0.05. Qualitative data from the focus group discussions underwent deductive thematic analysis using a six-phase protocol [32]. Two independent coders were involved in the analysis to ensure reliability and consistency in identifying themes. Intercoder reliability was established through regular meetings between the coders to compare and reconcile their interpretations of the data. Any discrepancies were resolved through discussion until consensus was reached. Trustworthiness of the findings was enhanced through member checking, where participants were invited to review and validate the identified themes to ensure they accurately reflected their perspectives. Additionally, triangulation was employed by comparing the qualitative findings with the quantitative results to corroborate findings and provide a more comprehensive understanding of the research phenomenon.

## 3. Results

Table 2 presents data on the demographics and characteristics of the participants in the study, comparing the treatment group (TG) and control group (CG). The initial numbers of participants in the TG and CG were 85 and 44, respectively. Of these, 58 participants in the TG and 36 in the CG completed the study, while 26 participants in the TG and 8 in the CG dropped out. The retention rates were 69% in the TG and 83% in the CG. (The Retention Rate was calculated as the percentage of participants who completed the study out of the initially assigned participants.) Among the completed participants in the TG, 43% were girls, and 56% were boys. The mean age of the completed participants in the TG was 15.4 years with a standard deviation of 0.89, while in the CG, the mean age was 15.0 years with a standard deviation of 1.00.

Table 3 presents the critical results of the intervention programme’s impact on the physical activity levels, anthropometry, and cardiorespiratory fitness. The results show that the intervention programme had a significant effect on the endline PA level (F (14, 79) = 4.884, *p* < 0.001), accounting for a substantial proportion of the variance (R squared = 0.464). Specifically, the baseline PA scores (F (1, 79) = 24.755, *p* < 0.001), intervention (F (1, 79) = 7.520, *p* = 0.008, η2 = 0.156), and interaction between intervention and age (F (2, 79) = 7.520, *p* = 0.008) demonstrated significant effects on the endline PA level scores. The intervention programme significantly affected the BMI (*p* = 0.025, partial eta squared = 0.025, η2 = 0.004), suggesting that the treatment led to changes in the BMI over time. Age exhibited a statistically significant effect on the BMI (*p* = 0.002, partial eta squared = 0.167), indicating a positive association with increasing age. Conversely, gender did not significantly influence the BMI (*p* = 0.545, partial eta squared = 0.005). Participation in the intervention programme significantly affected the WHR (*p* < 0.001, partial eta squared = 0.327, η2 = 0.100), suggesting that the treatment led to changes in the WHR over time. Waist-to-hip ratio (WHR) values of 0.80 (±0.10) in the CG and 0.81 (±0.09) in the TG were found, with no statistically significant difference between groups (*p* = 0.556). Age significantly affected the WHR (*p* < 0.003, partial eta squared = 0.163), with increasing age associated with higher WHR values. Gender did not significantly influence the WHR (*p* = 0.139, partial eta squared = 0.139), indicating similar WHR levels across genders. Participation in the intervention programme did not significantly affect the maximal oxygen uptake (VO_2_ max) (*p* = 0.102, partial eta squared = 0.033, η2 = 0.033), suggesting that there was a small but insignificant change in the VO_2_ max over time. The maximal oxygen uptake (VO_2_ max) values were 42.4 (±8.7) in the CG and 41.3 (±6.8) in the TG, with no statistically significant difference between groups (*p* = 0.396). Age (*p* < 0.711, partial eta squared = 0.017) and gender (*p* = 0.672, partial eta squared = 0.002) did not significantly influence the VO_2_ max, indicating similar VO_2_ max levels across different age groups and genders. 

Table 4 presents the key themes, subthemes, and verbatim quotes of the participants (*n* = 48) from the focus group discussion. Key themes and subthemes emerged regarding the behavioural outcomes of the intervention programme, which highlighted significant shifts in knowledge, risk perception, and attitudes towards exercise and health. The participants reported an enhanced understanding of the importance and benefits of physical activity, citing the Nyakaza campaign’s influence on their awareness.

## 4. Discussion

This study provides valuable insights into the efficacy of a multicomponent physical activity intervention programme (Nyakaza Move-for-Health) in addressing the pressing issue of sedentary lifestyle and insufficient physical activity among adolescents in under-resourced communities. The findings of this study show a statistically significant improvement in the physical activity levels, body mass index (BMI), waist-to-hip ratio, and predicted maximum oxygen uptake (VO_2_ max) of the adolescents who participated in a 12 week multicomponent physical activity intervention programme.

Despite the significant increase in the physical activity levels observed among the adolescents in the treatment group, the current study found that these levels still fell short of meeting the internationally recommended cut-off threshold of 2.75 proposed by Kowalski et al. [24]. This benchmark score indicates whether adolescents engage in sufficient physical activity to maintain their health. While the physical activity intervention programme resulted in significant improvements, the failure to meet this threshold demonstrates the persistence of challenges in promoting adequate physical activity levels among adolescents in under-resourced communities. The findings are consistent with international studies [33,34,35,36] which reported that adolescents struggle to meet recommended levels of physical activity, especially those from lower socioeconomic status countries and communities. In this study, the failure to meet the recommended levels of PA could be attributed to the decline in physical activity engagement in other domains, such as at school and at home. Therefore, it is imperative that an integrated approach to physical activity be employed to address physical inactivity and sedentary behaviour in all domains of leisure time, transport, home, and school [37].

The study results further revealed a significant decrease in the BMI over time, which suggests that the intervention contributed to changes in the body compositions of the participants. The intervention’s impact on body composition is crucial, considering the rise in obesity rates and associated health risks among adolescents [38,39]. The current research aligns with previous studies [40,41] which reported that multicomponent physical activity intervention programmes for adolescents resulted in significant but lower improvement in body composition measures such as the BMI and waist-to-hip ratio and a low body fat percentage for the treatment group compared with the control group. The current study intervention’s effect on the BMI accounted for modest changes, which suggests that the intervention alone may not be enough to combat obesity and its associated risks effectively. Longer and more intensive interventions are recommended for adolescents in order to realise significant changes in body composition [42,43]. Prolonged strategies in the treatment of obesity seem to be more effective compared with short-duration interventions [43].

This study’s examination of the waist-to-hip ratio (WHR) dynamics in response to intervention revealed compelling insights into the impact of targeted interventions on body composition and cardiovascular health. While both the treatment group (TG) and control group (CG) experienced changes in the WHR over the study period, the TG exhibited a more favorable outcome, suggesting the intervention’s effectiveness in modifying body fat distribution and central adiposity. These results support earlier studies [44,45] showing that a short-term physical activity intervention programme may produce a statistically significant decrease in the waist circumference (WC) and waist-to-hip ratio of adolescents. The substantial effect of the intervention on the WHR underscores its crucial role in promoting healthy body fat distribution and reducing cardiovascular disease risk [44]. These findings contribute to the growing body of evidence supporting the importance of tailored interventions in fostering favorable changes in body composition and overall cardiovascular health [45].

Regarding the intervention’s effect on the predicted maximum oxygen uptake (VO_2_ max), the results revealed that the treatment group showed a slight increase in the VO_2_ max, in contrast to the decrease in the VO_2_ max noted for the control group. Despite these trends, the intervention did not have a statistically significant effect on the VO_2_ max. This is contrary to a recent research study [46] which revealed that six weeks of circuit training significantly increased the predicted VO_2_ max of adolescents who participated in the intervention programme. Contrary to previous research studies [47,48,49], gender and age did not produce any significant influence on the VO_2_ max of the treatment group. Therefore, future research should consider larger sample sizes, longer intervention durations, and control for additional confounding variables to better understand the determinants of VO_2_ max changes in adolescents.

The examination of the psychosocial impact of the Nyakaza Move-for-Health intervention on adolescents revealed that the intervention programme sparked significant shifts in their attitudes towards physical activity and fitness. The findings of this research highlight the multi-faceted impact of the intervention programme. Underpinned in the transtheoretical model and health belief model, the intervention programme significantly enhanced participants’ awareness of the importance and benefits of physical activity. The results also show that the intervention increased knowledge about the dangers of physical inactivity and the positive impact of physical activity in adolescent’s health. Participant 4 mentioned that “before the Nyakaza campaign, I didn’t realise how dangerous sitting around all the time could be, posters and messages made me think twice about being inactive”. The findings are consistent with recent research studies which found that targeted health campaigns can effectively raise awareness and change health behaviours [50]. Participant 13 stated that “knowing about the benefits” and “messages on the exercise book covers changed how I perceive exercise”, highlighting how educational materials can shift perceptions about physical activity.

The research findings also show that the campaign heightened participants’ awareness of the personal risks associated with physical inactivity. Participant 4 stated that “the campaign opened my eyes to the risks of just sitting at my desk or on the couch all day. Now, I try to move more, even if it’s just a little bit every day”. The sentiments of Participant 3 were aligned with recent studies, stating that “awareness of health risks is a critical motivator for behaviour change” [50,51,52,53]. Participant 11 emphasised the campaign’s effectiveness, stating that “The Nyakaza campaign didn’t just talk about exercise, it showed us the real dangers of doing nothing. Now, I’m more aware, and I want to be active to stay healthy”.

The campaign positively influenced participants’ attitudes and beliefs about their capabilities for exercise. Initially, some participants held misconceptions about the physical effects of exercise, particularly regarding body image. Participant 23 mentioned, “In the beginning, we used to think that we would develop large calf muscles, which we thought would be unattractive in women, but as we learnt more about physical activity, we can now see that this is a fallacy and our body image is actually improved”. This shift in perception is crucial, as a negative body image can be a significant barrier to exercise [54,55,56]. Participant 18 discussed overcoming societal perceptions about women’s fitness: “We were raised with a perception that a woman who is fit and well defined is not pleasing to look at”, and “some of us fear to exercise because we will lose weight which is taken as unattractive in our society, but now I understand that I can exercise and still look like a woman”. Additionally, Participant 4 expressed newfound enjoyment in daily exercise, and Participant 3 shared their increased confidence and consistency in exercising.

The participants identified both extrinsic and intrinsic motivators that helped them overcome barriers to exercise. Extrinsic motivators included promotional materials and parental support. For instance, Participant 3 stated, “The watches played a huge role because of the readings or fitness markers it gave, and the messages on the activity diaries encouraged us to exercise”. Parental support was also crucial, as noted by Participant 17: “My grandmother would actually tell me when it was time for the programme”. Intrinsic motivators, such as enjoyment and fun, were also significant. Participant 11 highlighted the role of music in making exercise enjoyable: “Music was the best component because it made us not pay attention to the fact that we were exercising”. These findings are consistent with the literature, which underscores the importance of both intrinsic and extrinsic motivators in sustaining physical activity [57].

Our study showed that all participants demonstrated commendable commitment to the intervention programme, consistently engaging in and adhering to it throughout the 12 week period. This widespread participation suggests a collective transition to the action stage of change, indicating that individuals were actively striving to modify their behaviour and improve their physical activity levels. Furthermore, our study revealed that engagement in the programme was a deterrent to adolescent antisocial behaviours, as participants were occupied with extramural activities. This finding resonates with prior research emphasizing the role of physical activity in fostering friendships and social bonds among children and adolescents [58,59,60,61].

While this study provides important evidence supporting the efficacy of the Nyakaza Move-for-Health intervention, several limitations and areas for improvement are apparent. The findings suggest that while the intervention had positive effects, it may need to be more comprehensive, prolonged, and integrated across multiple life domains to achieve significant and lasting health benefits. Future research should address these limitations by considering larger sample sizes and longer durations and incorporating more robust measures of physical activity and fitness to build on these promising results.

## 5. Conclusions

This study underscores the potential effectiveness of the physical activity intervention programme (Nyakaza Move-for-Health) in addressing the prevalent issues of sedentary lifestyles and inadequate physical activity among adolescents. While future studies of a similar nature should look into the challenges of participant retention and more objective metrics of physical activity assessment, the findings are valuable in the context of under-resourced communities. Accordingly, the intervention resulted in statistically significant improvements in physical activity levels, body mass index, and waist-to-hip ratios, but it is imperative to acknowledge persistent challenges, such as participants failing to meet internationally recommended physical activity thresholds and variable responses based on gender and age. Additionally, despite slight improvements, concerns remain regarding the body mass index and cardiorespiratory fitness levels (VO_2_ max) of the participants. However, the intervention positively influenced body image satisfaction and attitudes towards physical activity and served as a deterrent to antisocial behaviours, highlighting its broader impact beyond physical health outcomes. The qualitative data shed light on the mechanisms through which the intervention programme may have influenced participants’ behaviours and attitudes towards health.

Overall, the integration of quantitative and qualitative findings provides a comprehensive understanding of the intervention programme’s impact on participants’ health and well-being. While the quantitative results offer objective measurements of changes in physical health indicators, the qualitative insights offer depth and context, helping to elucidate the mechanisms underlying these changes. Together, these findings contribute to the evaluation and refinement of health promotion interventions aimed at improving population health outcomes.

In the future, targeted interventions that account for gender differences and focus on enhancing the body mass index and cardiorespiratory fitness will be warranted to optimise the effectiveness of physical activity programmes among diverse adolescent populations, ultimately promoting holistic well-being and healthy lifestyle choices. This research significantly contributed to the limited knowledge regarding effective theoretical methods and strategies for promoting physical activity participation in this demographic. This study innovatively integrated various behavioural models, forming a new theoretical framework that not only predicts but also describes and explains the physical activity behaviour of adolescents, thereby advancing our understanding of the complex interplay between theoretical constructs and practical outcomes in this population.

## Figures and Tables

**Table 1 ijerph-21-00717-t001:** Intervention programme’s change objectives, performance objectives, methods, and practical application.

Change Objective (Outcomes)	Performance Objectives	Method	Practical Application
Improve knowledge of PA and its benefits.	Identify the importance and benefits of PA.	Social marketing: conscious-raising messages	Expose participants to the social marketing campaign infographics about the benefits of PA.
2. Raise awareness of the dangers of a sedentary lifestyle and physical inactivity.	Identify personal risks of physical inactivity.	Social marketing: conscious-raising messages	Expose participants to the social marketing campaign messages about the dangers of a sedentary lifestyle and physical inactivity.
3. Improve PA self-efficacy.	Feel confident about one’s capabilities to exercise.	Persuasive communication Empowerment	Facilitators provide positive feedback. Facilitators allow for mastery of experience (exercises).
4. Improve PA levels, body composition, and cardiorespiratory fitness.	60 min of moderate-to-vigorous physical activity (MVPA) per day. Meet body composition and cardiorespiratory fitness test norms.	Physical activity intervention programme	Moderate-to-vigorous physical activity intervention programme for 12 weeks for at-risk adolescents.

**Table 2 ijerph-21-00717-t002:** Demographics and characteristics of participants.

Groups	N (%)	Gender	N (%) per Group	Age Range Mean (SD) 15.4 (±0.89)	N (%)
Control Group (CG)	36 (38%)	Girls	41 (44%) CG 16 (44%) TG	13–14 Years	14 (15%)
Treatment Group (TG)	58 (62%)	Boys	53 (56%) CG 20 (56%) TG	15–16 Years	49 (52%)
				17 Years	23 (24%)

Note: TG = treatment group; CG = control group.

**Table 3 ijerph-21-00717-t003:** Physical activity levels, anthropometric measures and cardiorespiratory measures at baseline and endline.

Measurement	Groups	Baseline	Endline	Test Effects	F	*p* Value	Partial Eta Squared
		Mean (SD)	Mean (SD)				
*Physical Activity*	TG	2.2 (0.4)	2.6 (0.4)				
	CG	2.2 (0.4)	2.2 (0.5)				
				Baseline PA	24.755	<0.001	0.239
				Intervention	7.520	0.008	0.087
				Age * Intervention	7.520	0.008	0.039
				Int. * Gender	2.276	0.135	0.002
*Body Mass Index (BMI)*	TG	22.4 (4.6)	21.9 (4.2)				
	CG	21.1 (3.3)	21.0 (2.6)				
				Baseline BMI	177.930	<0.001	0.693
				Age	5.276	0.002	0.167
				Gender	0.370	0.545	0.005
				Intervention	2.059	0.025	0.025
*Waist-to-Hip Ratio*	TG	0.80 (0.10)	0.76 (0.9)				
	CG	0.81 (0.09)	0.82(0.07)				
				Baseline WHR	98.939	<0.001	0.556
				Age	5.116	<0.003	0.163
				Gender	12.773	<0.001	0.139
				Intervention	38.458	<0.001	0.327
*Maximal Oxygen Uptake (VO_2_ max)*	TG	42.4 (8.7)	43.6 (7.7)				
	CG	41.3 (6.8)	40.9 (5.9)				
				Baseline VO_2_ max	51.777	<0.001	0.396
				Age	0.461	0.711	0.017
				Gender	0.180	0.672	0.002
				Intervention	2.736	0.102	0.033

Note: TG = treatment group; CG = control group. SD = standard deviation. The *p* value indicates statistical significance, with significance denoted by the body mass index (BMI). WHR = waist-to-hip ratio; VO_2_ max = maximal oxygen uptake. * *p* < 0.05.

**Table 4 ijerph-21-00717-t004:** Behavioural outcomes of intervention programme.

Key Themes	Subthemes	Verbatim Quotes
(1) Knowledge	Importance and benefits of PA	Participant 4: “Before the Nyakaza campaign, I didn’t realise how dangerous sitting around all the time could be. The posters and messages made me think twice about being inactive.”
		Participant 17: “Knowing about the benefits” “messages on the exercise book covers changed how I perceive exercise”
		Participant 12: “Socialising with other people was another key benefit of programme,”
		Participant 9: “When you don’t exercise, you are always at home, and you easily become involved in substance abuse and social ills because there is no extramural activity that you do.”
(2) Risk perception	Personal risks of physical inactivity	Participant 4: “The campaign opened my eyes to the risks of just sitting at my desk or on the couch all day. Now, I try to move more, even if it’s just a little bit every day.”
		Participant 3: “The messages on the t-shirts were like a wake-up call. I never thought sitting too much could harm me, but now I see the risks. It’s a motivation to keep moving.”
		Participant 11: “The Nyakaza campaign didn’t just talk about exercise; it showed us the real dangers of doing nothing. Now I’m more aware, and I want to be active to stay healthy.”
(3) Attitudes and beliefs	Confident about one’s capabilities to exercise	Participant 23: “in the beginning, we used to think that we would develop large calf muscles, which we thought would be unattractive in women, but as we learnt more about physical activity, we can now see that this is a fallacy and our body image is actually improved.”
		Participant 18: “We were raised with a perception that a woman who is fit and well defined is not pleasing to look at” and “some of us fear to exercise because we will lose weight which is taken as unattractive in our society, but now I understand that I can exercise and still look like a women”
		Participant 4: “I never thought I could enjoy exercising every day, but this program made it fun. Now, I look forward to it!”
		Participant 3: “Being consistent with daily exercise wasn’t easy at first, but now I feel like it’s a part of who I am. I have more confidence in my abilities.”
(4) Barriers and motivation	Extrinsic motivators:	
	Promotional materials Parental support	Participant 3: “The watches played a huge role because of the readings or fitness markers it gave, and the messages on the activity diaries encouraged us to exercise.”
	Intrinsic motivators:	Participant 17: “My grandmother would actually tell me when it was time for the programme.”
	Enjoyment and fun Music enhances the overall experience	Participant 11: “Music was the best component because it made us not pay attention to the fact that we were exercising.”

## Data Availability

The data presented in this study are available on request from the corresponding author. The data are not publicly available due to data privacy and ethical concerns.

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
