# Peer review of "Impact of a Nyakaza Move-for-Health Intervention Programme among Adolescents in a Resource-Constrained South African Community"

_ijerph, 2024, doi:10.3390/ijerph21060717_

Round 1
Reviewer 1 Report
Comments and Suggestions for Authors
This manuscript was very easy and interesting to read. My only comment is that the paragraph about Table 6 seems to be on page 7 and 8, so one of those should be deleted.
Reviewer 2 Report
Comments and Suggestions for Authors
This study evaluates the impact of a 12-week Nyakaza move-for-health intervention on physical activity, anthropometry, cardiorespiratory fitness, and behavior change among adolescents in KwaZulu-Natal, South Africa.
I consider the subject to be interesting and important. However, I have some suggestions.
General suggestions
- Line 123, I suggest that you do not repeat yourself in the sentence: "to engage in physical activity actively".
- I suggest better specifying what kind of procedure was performed with the control group participants. It is unclear whether or not they were exposed to the campaign process in their school, or how the researchers selected the students who participated in the control or experimental group.
- It is unclear why the authors did not indicate the gender distribution of the control group in Table 2.
- I believe that the distribution and presentation of the data in Table 2 could be improved. See APA recommendations for tables.
- Although the authors found no differences in PA level in the control group, it is important to describe the data in Table 4.
- It is important that the authors describe the number of students in age groups. Depending on this, also describe the statistical analysis (it probably corresponds to a non-parametric test instead of Student's t-test).
- The authors never mentioned the "watches" in the procedure, although they are mentioned in the qualitative testimonies. It is important to accurately describe all the instruments and materials used.
- In lines 223-229 a previous paragraph is repeated.
- Check the quotation in line 239.
- In the discussion, the argument of the differences observed in the girls in terms of BMI mean could also include the cultural factor: "...the perception that a woman who is fit and well defined is not pleasing to look at” and “some of us fear to exercise because we will lose weight, which is taken as unattractive in our society, but now I understand that I can exercise and still look like a women”.
- The authors evaluate the action stage of change, but did not present results of this type in the groups and what happened after the intervention.
Comments on the Quality of English LanguageMinor editing of English language required.
Reviewer 3 Report
Comments and Suggestions for Authors
Thank you for the opportunity to review your study. This is important work that contributes to our understanding of best practices for youth physical activity and health promotion in a resource-constrained South African community. I have a few suggestions that I hope you find useful:
Abstract
Line15-16. Please consider including mean scores, SD, and test statistics. Including only p-values and effect sizes is not sufficient.
Line 17. Did the authors conduct any post-hoc power analysis on the non-significant results?
Introduction
The background of the 'problem' has been well documented. However, more details are needed in this section. Consider including literature related to the big picture, such as ecological frameworks and youth physical activity and health, as physical activity in youth is complicated and related to numerous factors. Please revise this section to include a theory or theoretical framework that is fundamental to your work. In addition, do the authors have any research questions or aims that they want to address at the end of the introduction section?
Methods
Line 67. The quasi-experimental design seems to include "pre-post-test", which is likely redundant here. Please consider removing this for clarity.
Line 80-81. Is there a specific reason for the imbalanced group assignment?
Line 83-94. Please consider including detailed information regarding the tests (e.g., psychometric properties, sample items from scales, validity, reliability, devices used for cardiorespiratory fitness) and the reason and criteria for selecting these measurements for this study.
Line 99. What does "qualified" mean, and what is the definition of "qualified"?
Line 105-106. These models need to be introduced in the introduction section.
Line 108. Please define "tailored messages".
Table 1. Please clarify whether the objectives, method, outcomes, and applications are only for the experimental group or for both groups in different sequences.
Line 147. Please define "applied sequentially (QUAN + QUAL)" and justify why the authors did this "sequentially".
Line 149. Using t-test is not appropriate for this type of design. Given that the authors tested two groups at two time points, factorial ANOVA should be the correct statistical approach. Otherwise, there will be severe type-I error inflation. This is one of the major concerns of this study.
Line 151. Cohen's d should be reconsidered for ANOVA. Partial eta-squared is recommended.
Line 154-157. Need to provide details regarding the thematic analysis, including the number of coders, intercoder reliability, and how the authors established trustworthiness and triangulation. The method section needs to be sufficiently revised with details and accurate information, and data needs to be rerun through appropriate statistical analysis.
Results and discussions
Table 2 needs to be revised for clarity; please consider revising row and column titles.
Lines 171-176: F test results and effect sizes will need to be added to the result section throughout after the authors rerun their analysis.
Given that the methods applied in the current study are questionable, the results and discussion sections need to be revised based on the updated results. Additionally, did the authors run a power analysis to determine the sample size? I highly recommend that the authors run a post-hoc power analysis regarding the non-significant results to help reviewers/readers understand whether this is a sample size issue or if there is not much difference between the CG and EG.
Reviewer 4 Report
Comments and Suggestions for Authors
I hope this letter finds you well. I had the opportunity to review your article titled, “Impact of a Nyakaza Move-for-Health Intervention Programme among Adolescents in a Resource-Constrained South African Community”, which was submitted to International Journal of Environmental Research and Public Health.
Abstract
The abstract is a summary of the study. It is believed that the abstract of this study summarizes each part of the study well.
Introduction
The need for research is well explained. In the introduction, prior research was appropriately presented to emphasize the need for research. However, it is judged that it is difficult for readers to understand the content because the sentences are written in long sentences. Please revise the introductory sentence to a short sentence to convey the content accurately.
Materials and Methods.
Research methods, procedures, procedures, and statistical analysis were well described. The research design is judged to be very good. And the application of mixed methods was very impressive. In particular, the research methods of social marketing campaigns and afterschool PA sessions are considered very interesting. It is believed that this is a design that allows for accurate interpretation and understanding of the results of this study.
Results
It is judged that the results necessary for this study were derived through the research process.
Discussion
In the discussion, the results of this study and previous research are faithfully analyzed and presented. However, the researcher's argument is judged to be somewhat lacking. There is a need to logically explain the researcher's claims regarding the purpose of the study.
[6] warned that low levels of cardiorespiratory fitness are associ-257 ated with increased clustering of metabolic abnormalities of the syndrome. [28] associated 258 low cardiorespiratory fitness values with overweight, obesity, hypertension, and hyper-259 triglyceridemia, emphasizing the importance of addressing cardiorespiratory fitness in 260 adolescent health interventions.
The subject of the above sentence is omitted.
Conclusions
I think the conclusion is well written. Suggestions for follow-up research are included. This study is expected to be used as good data for youth in South Africa.
Comments on the Quality of English Language
Minor editing of English language required.
Round 2
Reviewer 3 Report
Comments and Suggestions for Authors
Lines 194-195: Please consider reporting alpha instead of the p.
Table 2 can be improved and reformatted. In the third row, the "%" symbol is missing. Also, please clarify if the value 0.89 represents the SD.
Thank you for reporting partial eta squared. However, I noticed that authors reported both eta squared and partial eta squared. Is there a specific reason for reporting both effect sizes simultaneously?
While Table 4 offers a succinct overview of the results, it may not fully capitalize on the richness of the qualitative data. Given that mixed-method design is a strength of this study, I recommend elaborating on the qualitative findings in the discussion section. This will provide a more comprehensive understanding of the results and their implications, rather than primarily focusing on the quantitative outcomes.
I acknowledge and appreciate the authors' efforts in addressing the previous revisions. However, I believe that the methodological approach employed in the current study warrants further scrutiny. Specifically, the authors should provide more detailed justifications for the chosen methods and discuss their limitations. Furthermore, the results and discussion sections would benefit from revision to enhance clarity and coherence. In the discussion, I recommend focusing on the interpretation of the findings and their implications, while minimizing the use of statistical jargon (e.g., p-values and effect sizes). To improve the overall quality of the manuscript, I encourage the authors to carefully review the scientific writing, ensuring consistency and clarity in the reporting and interpretation of the results.
